# Comparison of Short-Term Effects of Different Spinal Manipulations in Patients with Chronic Non-Specific Neck Pain: A Randomized Controlled Trial

**DOI:** 10.3390/healthcare12131348

**Published:** 2024-07-05

**Authors:** Jessica García-González, Raúl Romero-del Rey, Virginia Martínez-Martín, Mar Requena-Mullor, Raquel Alarcón-Rodríguez

**Affiliations:** Department of Nursing, Physiotherapy, and Medicine, Faculty of Health Sciences, University of Almería, 04120 Almería, Spain; jgg145@ual.es (J.G.-G.); mrm047@ual.es (M.R.-M.); ralarcon@ual.es (R.A.-R.)

**Keywords:** disability evaluation, musculoskeletal manipulations, neck pain, spinal manipulation

## Abstract

Spinal manipulations for chronic non-specific neck pain (CNNP) include cervical, cervicothoracic junction, and thoracic spine (CCT) manipulations as well as upper cervical spine (UCS) manipulations. This study aimed to compare the short-term effects of UCS manipulation versus a combination of CCT spine manipulations on pain intensity, disability, and cervical range of motion (CROM) in CNNP patients. In a private physiotherapy clinic, 186 participants with CNNP were randomly assigned to either the UCS (n = 93) or CCT (n = 93) manipulation groups. Neck pain, disability, and CROM were measured before and one week after the intervention. No significant differences were found between the groups regarding pain intensity and CROM. However, there was a statistically significant difference in neck disability, with the CCT group showing a slightly greater decrease (CCT: 16.9 ± 3.8 vs. UCS: 19.5 ± 6.8; *p* = 0.01). The findings suggest that a combination of manipulations in the CCT spine results in a slightly more pronounced decrease in self-perceived disability compared to UCS manipulation in patients with CNNP after one week. However, no statistically significant differences were observed between the groups in terms of pain intensity or CROM.

## 1. Introduction

Neck pain represents a significant public health problem in the general population [1], leading to considerable economic costs for the healthcare system [2]. It is considered one of the leading causes of disability in most countries [3], with its prevalence varying widely across populations and age groups [1,4]. A recent study reported an increase in the number of cases from 164.3 million in 1990 to 288.7 million in 2017, with Western Europe being one of the regions with higher prevalence [1]. Given the rising number of cases in recent years and the negative impacts it poses on both personal and societal levels in terms of quality of life and work capacity, there is a call for neck pain to be prioritized in future research concerning prevention and treatment [5,6].

Non-specific neck pain is defined as a subjective sensation of discomfort in the cervical region of the spinal column, aggravated by movements and/or prolonged postures. It is important to note that this category of neck pain excludes cases associated with neurological deficits, vascular deficits, neoplasms, or infections [7]. Most of these patients may experience a limited range of motion in the cervical spine, as well as pain in the shoulders and neck, which can be associated with limitations in activities of daily living and disability [8]. It is estimated that 50% of these patients will continue to experience some symptoms one year after their first onset, thereby becoming a chronic problem [9]. In this regard, there is scientific evidence supporting the importance of physiotherapy in the treatment of these patients [6].

Despite being recommended by Clinical Practice Guidelines, cervical spinal manipulation is considered an adjunct to education and exercise therapy within the framework of a multimodal care approach for patients with chronic non-specific neck pain (CNNP) [10,11]. These techniques appear to exert their effects not specifically on a single vertebral joint but on multiple joints; these effects could be linked to biomechanical interactions and systemic effects, such as alterations in the functioning of the descending antinociceptive system and central mechanisms of pain modulation [12]. Moreover, a recent systematic review and meta-analysis reported that physiotherapy interventions for musculoskeletal pain, including spinal manipulations, produce benefits influenced by contextual factors not attributable to the specific effects of the interventions. Furthermore, the deliberate use of these contextual factors to enhance therapeutic outcomes was recommended, representing an ethical opportunity with potential benefits for patients [13].

Previous studies recommend the inclusion of cervical and thoracic spinal manipulation into that multimodal therapeutic approach to achieve short-term pain reduction and improve both function and cervical range of motion (CROM) in addressing non-specific cervical pain [14,15,16]. However, a recent systematic review and meta-analysis has reported on the effectiveness of spinal manipulations alone in reducing neck pain and disability in patients with CNNP, highlighting the need for more randomized clinical trials due to the limited number of studies available to date [17]. Specifically, a previous study observed a reduction in both neck pain and disability in patients who underwent manipulations of the mid-cervical spine, as well as in those who received a combination of manipulations involving the cervical, cervicothoracic junction, and thoracic spine (CCT), with disability outcomes being superior in the latter group [18]. Given the inherent biomechanical connection between the thoracic and cervical spine, disruptions in thoracic spine biomechanics may significantly contribute to the onset of neck pain [19]. Alternatively, the zygapophyseal joints of the cervical spine, particularly the upper cervical joints, are densely innervated. It has been proposed that dysfunction in the upper cervical spine (UCS) could compromise the normal function of mechanoreceptors, affecting the soft tissues that respond to and generate movements in the neck [20]. Recent evidence suggests that the treatment of the UCS improves symptoms in these patients [21,22,23], including an enhancement of the range of motion in the UCS [24], pressure pain threshold in various cervical muscles [25], pain, and self-perceived disability [17,21,26]. However, to date, there are no studies comparing the combination of various spinal manipulations in the CCT spine with UCS manipulation, which could assist healthcare professionals in making treatment decisions for these patients. Therefore, this trial aimed to compare the short-term effects of UCS manipulation versus a combination of CCT spine manipulations on pain intensity, disability, and CROM in patients suffering from CNNP.

## 2. Materials and Methods

### 2.1. Research Design and Ethics

This study was a single-blind, parallel, randomized controlled trial in which the outcome assessor was blinded to the treatment allocation. Due to the nature of the study, neither the researcher conducting the intervention nor the participants were blinded to the treatment allocation. However, the participants were blinded to the study’s objective. The statistician was also blinded to the treatment allocation. The study was approved by the Ethics and Research Commission of the Department of Nursing, Physiotherapy and Medicine at the University of Almería (Registration No. EFM 81). It was registered online on ClinicalTrials.gov (NCT04268667) in accordance with all Consolidated Standards of Reporting Trials guidelines [27].

### 2.2. Participants

Participants with CNNP were recruited from February 2023 to March 2024 through posters and social media. Those who saw the study advertisement visited a private physiotherapy clinic to participate where the study was conducted. The study adhered to the Declaration of Helsinki, and each participant signed an informed consent form before their enrolment in the study. The inclusion criteria were as follows: (1) experiencing symptoms that persisted for over 12 weeks; (2) being 18 years of age or older; (3) experiencing localized pain in the cervical spine; and (4) having symptoms caused by cervical movements or prolonged postures. Participants were screened for signs of vertebrobasilar insufficiency, such as nystagmus, gait disturbances, or Horner’s syndrome [28]. Additionally, they were assessed for instability in the upper cervical spine ligaments using the Sharp–Purser test, alar ligament stress test, and transverse ligament tests [29]. Conversely, the exclusion criteria included (1) acute symptom onset; (2) contraindications to cervical spinal manipulation such as fractures, osteoporosis, joint infections, or vertebrobasilar insufficiency [30]; (3) a history of neck trauma or cervical spine surgery [30]; (4) a diagnosis of cervical radiculopathy; (5) a diagnosis of fibromyalgia; (6) a receipt of physiotherapy treatment within the last 3 months; and (7) a receipt of any other forms of treatment throughout the study.

### 2.3. Data Collection

The study was conducted at a private physiotherapy clinic and proceeded as follows: initially, participants provided baseline demographic information and outcome measures, including pain intensity, neck disability, and CROM, during a face-to-face interview with a physical therapist in a designated area. Subsequent to the baseline data collection, participants were randomly assigned to one of two study groups by drawing numbered tickets from a container overseen by the intervention therapist in a separate area. Immediately after randomization, the intervention therapist, who could not be blinded due to the nature of the study, administered the treatment corresponding to the assigned group. Upon the completion of the intervention, participants exited the study premises. One week after the intervention, the same physical therapist who conducted the initial assessment reassessed pain intensity, neck disability, and CROM.

### 2.4. Interventions

All interventions were carried out by a physical therapist with over 10 years of experience in spinal manipulation treatments, distinct from the one who collected the outcome measures. Participants received their respective treatments on the same day as their initial evaluation. According to a recent systematic review [12], all spinal manipulation techniques were performed uniformly, regardless of the patient’s pain location in the cervical spine. The interventions were executed as follows.

#### 2.4.1. Upper Cervical Spine Manipulation Group (C1–C2)

The participant was placed in a supine position, and the physiotherapist’s second finger was placed over the posterior arch of the atlas. A posterior–anterior shift, ipsilateral tilt, and contralateral side shift were performed until tension was felt on the finger. Finally, a single high-velocity, low-amplitude thrust manipulation was directed upward and medially in the direction of the patient’s contralateral eye [23] (Figure 1). This technique was applied bilaterally.

#### 2.4.2. Cervical, Cervicothoracic Junction, and Thoracic Spine Manipulations Group

All participants included in this group received three different spinal manipulations (Figure 2):

First, cervical spine manipulation (C3–C4) was performed with the participant in a supine position and the cervical spine in a neutral position. The physiotherapist’s second finger contacted the posterior–lateral aspect of C3’s zygapophyseal joint while holding the patient’s face with the other hand. An ipsilateral tilt and contralateral rotation were performed until tension was felt on the finger. A high-velocity, low-amplitude thrust manipulation was directed upward and medially in the direction of the patient’s contralateral eye [18]. This technique was applied bilaterally.

Second, thoracic spine manipulation (T5–T6) was applied with the patient in a supine position with arms crossed over the chest and hands on the shoulders. The physiotherapist placed their chest on the patient’s elbows, and the manipulative hand contacted the sixth thoracic vertebra. A thoracic flexion was introduced until slight tension was felt on the hand contact. Afterward, a high-velocity, low-amplitude thrust was applied, with the therapist’s weight lightly applied to the patient. This involved the simultaneous use of an antero-posterior and upward-directed component [18].

Lastly, cervicothoracic junction manipulation (C7–T1) was performed. For a right C7–T1 thrust joint manipulation, the patient was in a prone position with the head turned to the right and the physiotherapist to the right of the participant. The therapist’s left thumb contacted T1’s spinous process, and the right hand supported the patient’s head. A slight left tilt was performed until the thumb in contact felt tension. Lastly, a high-velocity, low-amplitude thrust was applied toward the patient’s right side [18]. This technique was applied bilaterally.

### 2.5. Outcome Measures

Demographic and clinical data, including gender, age, weight, height, pain location, and duration of pain, were gathered from participants. Subsequently, various baseline measurements of primary and secondary outcome measures were conducted, both at baseline and one week after the intervention, following the methodology of a previous study [18].

#### 2.5.1. Primary Outcome Measures

Pain intensity was assessed using the Numerical Pain Rating Scale (NPRS) [31]. This scale ranges from integers 0 to 10, where 0 represents “no pain” and 10 represents “the worst pain imaginable”. Participants selected the single number that best reflected their current pain level at rest. The minimum detectable change (MDC) for NPRS in patients with neck pain is 2.6 points [32].

Neck disability was measured using the Neck Disability Index (NDI), consisting of 10 items related to the patient’s daily activities, each scored from 0 (no limitation) to 5 (limitation or inability to perform that function). The total score is calculated by summing all items, then dividing by 50 (the maximum score) and multiplying by 100 to obtain a percentage. A score of 0% represents the highest level of independence, while 100% represents total dependence. This trial used an adapted Spanish version of NDI (α: 0,94) [33].

#### 2.5.2. Secondary Outcome Measures

CROM was measured using a reliable and valid CROM device (ICC 0.89 to 0.98) [34]. The device actively measured all cervical movements: flexion, extension, right and left lateral flexion, and right and left rotation. The CROM device was placed on the participants’ heads, and a magnetic collar, also part of the CROM device, was positioned on their shoulders to account for any trunk rotation. The collar consistently maintained the same position relative to the magnetic pole. The chair in which participants sat remained in a fixed position throughout the entire data collection. The initial head position was set to neutral at the inclinometer’s zero mark for flexion, extension, and both lateral flexions. For rotation, the dial was adjusted to zero with the head in a neutral position. When the movement occurred in one direction, the final position was read and recorded for each trial. Participants returned their heads to the initial position once the reading was completed. Each movement was performed three times, and the average was recorded. The measurement error for the CROM device was estimated to be between 1.6° and 2.8°, and the MDC was 6.5° [34].

### 2.6. Sample Size

The sample size was calculated using the Ene 3.0 software to estimate mean differences of 2.1 points in NPRS scores, with a standard deviation of 2.5 and a significance level of 5% [35]. To achieve a statistical power of 80%, the estimated sample size was 93 participants in both groups.

### 2.7. Randomization

A statistician utilized the online Research Randomizer tool [36], www.randomizer.org (accessed on 1 August 2019) prior to commencing the study to randomly allocate the 186 participants into the two groups. Codes were generated and randomly distributed between the groups, then placed in a box that was kept by the physiotherapist in charge of applying the interventions. Participants picked a number from the box to determine their assigned group: the UCS manipulation group or the CCT spinal manipulation group.

### 2.8. Data Analysis

The data were analyzed using IBM SPSS Statistics 23.0 [37]. An intention-to-treat (ITT) analysis was performed, where the analysis of the clinical trial participants was based on the group to which they were initially assigned and not on the treatment they finally received. A descriptive analysis was performed for continuous variables with means and standard deviations (means ± SD). The Kolmogorov–Smirnov test was used to determine the normality of the continuous variables. 

Baseline demographic and clinical variables were compared between groups using the Mann–Whitney U test for continuous data and Chi-square tests (X2) for categorical data. The values for pain intensity, neck disability index, and active CROM were expressed as means ± SD at baseline and one week, and as means (95% CI) for within-group change scores.

The variables analyzed (pain intensity, disability, and CROM) were non-normality distributed (*p* < 0.05). Within-group comparisons were made using the Wilcoxon test, and effect sizes were determined using Rosenthal’s r. Between-group comparisons used the Mann–Whitney U test for the nonparametric variable. A *p*-value of <0.05 was considered statistically significant.

## 3. Results

Initially, 212 patients with CNNP were recruited, of whom 26 were excluded for various reasons. Finally, a total of 186 participants were randomized to the UCS manipulation group (n = 93) and the CCT spine manipulation group (n = 93) (Figure 3).

The baseline characteristics of both groups demonstrated homogeneity regarding age, gender, height, weight, and pain location, revealing no statistically significant differences (as depicted in Table 1).

Table 2 presents baseline and post-intervention data, as well as between-group comparisons for pain intensity, disability, and CROM. Initially, the outcome measures’ characteristics were comparable across both groups, with no statistically significant differences observed. In relation to pain intensity, both groups experienced a decrease in the mean score one week after the intervention, with no statistically significant differences observed between the groups (*p* = 0.31). Regarding neck disability, a statistically significant difference was detected between the groups (*p* = 0.01), with the CCT manipulation group exhibiting lower scores (16.9 ± 3.8) compared to the UCS manipulation group (19.5 ± 6.8), reflected in a Rosenthal’s effect size of −0.54. Finally, no statistically significant differences were found between the groups for CROM one week after the intervention (*p* > 0.05), except for the left lateral flexion movement (CCT: 47.5 ± 8.8 vs. UCS: 43.9 ± 9.0, Rosenthal’s effect size: 0.40; *p* = 0.03).

When comparing these measures between both manipulation groups by sex, no statistically significant differences were found among men or among women.

Appendix A display intragroup results for the UCS manipulation group and the CCT manipulation group, respectively. They encompass mean values at baseline, one week post-intervention, and within-group change scores, alongside the associated 95% confidence interval (CI) and effect size for measures of pain intensity, disability, and CROM.

## 4. Discussion

This study aimed to compare the short-term effects of UCS manipulation versus a combination of CCT spine manipulations on pain intensity, disability, and CROM in patients suffering from CNNP. The results indicated a reduction in pain intensity in both groups, with no statistically significant differences observed between them. A decrease in neck disability was recorded for all participants, with statistically significant differences between the groups favoring those who received spinal manipulations in the CCT spine, showing slightly superior results. However, no improvement in CROM was observed in either group, as it did not reach the threshold established by the MDC for the measurement instrument.

The findings revealed a reduction in neck pain among participants from both groups one week after receiving their respective interventions, surpassing the MCID [32]. However, this pain reduction was comparable between groups, with no statistically significant differences observed among them. These findings are supported by previous research [18,38]. Martinez-Segura et al. [38] found that patients with CNNP who underwent spinal manipulations in both the cervical and thoracic spine experienced an immediate decrease in the intensity of resting pain, without statistically significant differences when comparing both groups. Similarly, Saavedra et al. [18] reported results consistent with this study regarding the intensity of cervical pain, observing a decrease in mean scores one week after applying spinal manipulations in the middle cervical spine or CCT spine, exceeding the MCID. No statistically significant differences were found in the intergroup comparison.

Regarding neck disability, participants in both groups showed moderate levels of disability at baseline (ranging from 30% to 48%) [39]. Initially, the disability level was 44.6% in the UCS manipulation group and 45.4% in the CCT spine manipulation group. After the interventions, disability levels decreased to 38.9% and 33.9%, respectively, leading to a mean decrease of 5.7% in the UCS spinal manipulation group and 11.5% in the CCT spinal manipulation group, with significant within-group and between-group differences. In the present study, the mean decrease in the CCT spine manipulation group exceeded the MCID of 10% variation [40]. However, the slight difference between groups did not surpass this MCID. Other authors have also compared the effectiveness of different spinal manipulations on disability in similar populations. Like this trial, Dunning et al. [41] found a mean decrease that exceeded the MCID two days after applying UCS and thoracic spinal manipulations. Saavedra et al. [18] also observed similar results seven days after the interventions, exceeding the MCID. Statistically significant differences were observed between groups, with better outcomes obtained by the group that received CCT spine manipulations. 

CROM increased in both groups in most ranges, with statistically significant results, but the mean difference did not exceed the established MDC of 6.5° [34]. This indicates that the techniques applied did not contribute to the variation. Consistent with these findings, several authors [18,38,42] also showed an increase in CROM with cervical and thoracic manipulations, but the change did not surpass the MDC. Considering these results, one session of spinal manipulation may not be enough to improve CROM in patients with CNNP after one week post-intervention. The lack of immediate post-intervention measurements prevents determining whether there was an immediate increase in CROM after spinal manipulations and whether it normalized over a number of days. Further research with multiple sessions over a longer period may reveal greater clinical relevance. 

Although an improvement in CROM was not observed, participants in both groups experienced a reduction in pain intensity with no differences observed between groups, suggesting that this effect may be more closely linked to the neurophysiological aspects of spinal manipulation than its mechanical effects [43]. However, the only outcome measure that revealed disparities between the groups was self-perceived disability. Patient self-perception of change is deemed crucial in patient management [44,45], particularly in the context of spinal manipulative therapy, whose neurophysiological mechanisms are not yet fully understood [46]. Thus, the observed effects one week after a single session of spinal manipulation in the current study may result from a combination of biomechanical, neurophysiological, and contextual factors [47]. Contextual factors such as patient expectations and response to the therapeutic alliance may exert a significant impact on clinical outcomes for individuals with neck pain [13,48,49,50]. These nonspecific factors could play a role in a more pronounced improvement following spinal manipulations. In this regard, participants assigned to the CCT spinal manipulation group were exposed to a broader set of spinal manipulation techniques compared to those in the UCS manipulation group. Consequently, these participants may have developed an enhanced perception of treatment and expectations, potentially influencing the assessment of their self-perceived disability after the intervention. Therefore, these findings should be interpreted with caution, although they can guide healthcare professionals in making treatment decisions for patients with CNNP, considering the inclusion of spinal manipulation techniques targeting the CCT spine in their treatment approach for these patients.

### Strengths and Study Limitations

The major strength of this study includes a large number of participants in both groups, which enhances the statistical power of the findings. However, the study presents some limitations: First was the absence of a placebo group and blinding due to the nature of the intervention. Second, participants in the CCT spinal manipulation group received a greater number of spinal manipulations than the UCS manipulation group, so it cannot be firmly asserted that the observed disparities between the groups regarding neck disability were not linked to contextual factors, primarily associated with participants’ expectations. Finally, the possibility of natural remission cannot be ruled out. These expectations, derived from previous experiences or beliefs about the efficacy of the intervention, could have affected the reported outcomes. Additionally, the therapist’s experience may have influenced the application of the spinal manipulation techniques and, consequently, the results. Finally, the possibility of natural remission cannot be ruled out. Future research should consider multiple sessions of spinal manipulations and therapeutic exercise and a longer-term evaluation of outcomes beyond the seven-day observation period in this study. 

## 5. Conclusions

In conclusion, a combination of manipulations in the CCT spine results in a slightly more pronounced decrease in self-perceived disability compared to UCS manipulations in patients with CNNP one week post-intervention. However, no statistically significant differences were observed between the groups in terms of pain intensity or CROM. These differences in self-perceived disability may guide healthcare professionals towards considering CCT spine manipulations for treating these patients, potentially due to providing more favorable contextual factors, including heightened treatment expectations among these patients.

For future research, although it is recognized that contextual factors can influence the short-term effects of spinal manipulation techniques, it is suggested that these effects could be enhanced by combining such techniques with exercise and education in the treatment of patients with CNNP in addition to extending the follow-up period to assess their long-term effects.

## Figures and Tables

**Figure 1 healthcare-12-01348-f001:**
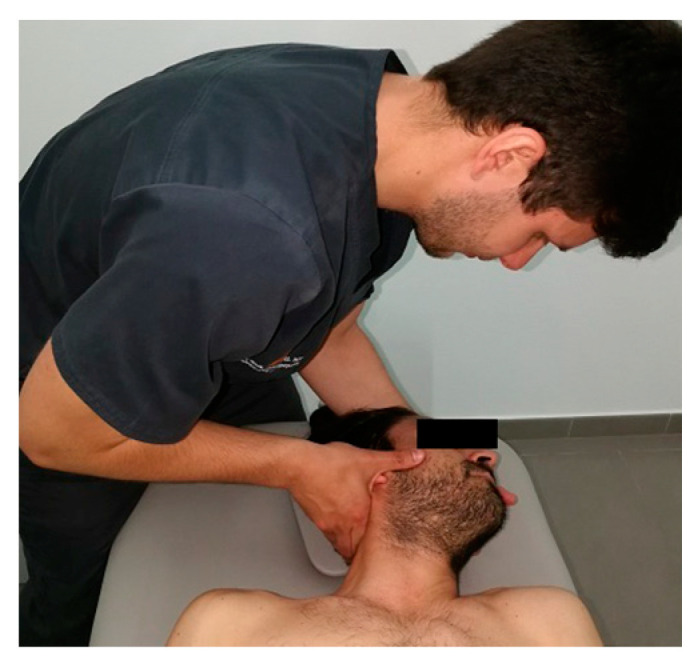
Atlantoaxial joint manipulation technique applied in the UCS group.

**Figure 2 healthcare-12-01348-f002:**
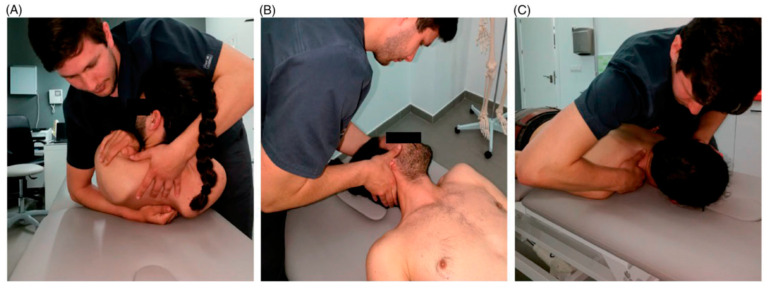
Spinal thrust joint manipulation techniques applied in the CCT spine manipulation group. (**A**) Thoracic spine manipulation; (**B**) Mid-cervical spine manipulation; (**C**) Cervico-thoracic junction manipulation.

**Figure 3 healthcare-12-01348-f003:**
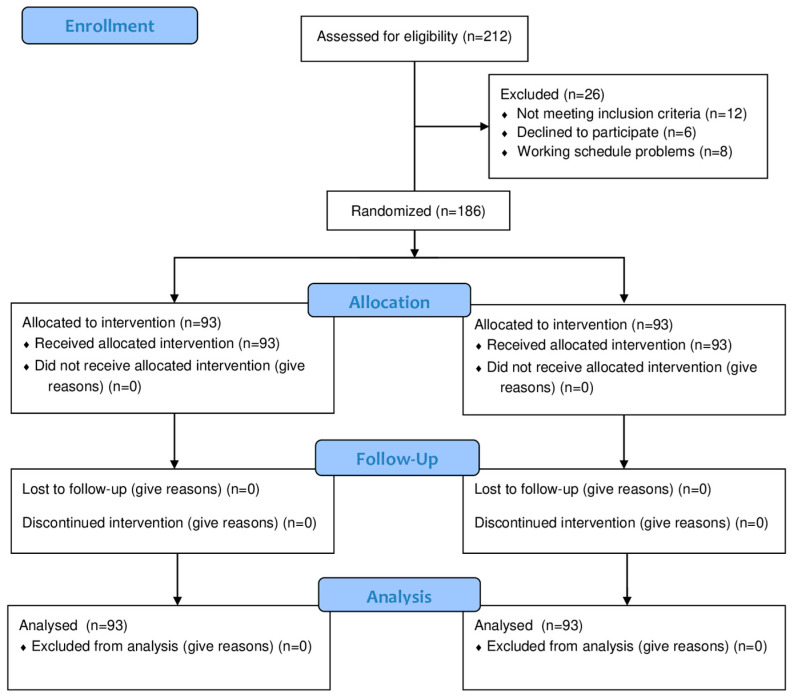
Flowchart Diagram According to the CONSORT Statement for the Report of Randomized Controlled Trials.

**Table 1 healthcare-12-01348-t001:** Physical and clinical baseline characteristics of the sample.

	CCT Manipulations Group (n = 93)	UCS Manipulation Group (n = 93)	*p*-Value
Male	38 (40.9%)	29 (31.2%)	0.16 ^a^
Female	55 (59.1%)	64 (68.8%)	
Age (years)	31.9 ± 9.7	33.5 ± 10.7	0.28 ^b^
Weight (kg)	70.8 ± 14.3	68.6 ± 14.5	0.30 ^b^
Height (cm)	170.8 ± 11.6	170.0 ± 9.2	0.61 ^b^
Duration of pain (years)	2.2 ± 0.87	2.3 ± 0.64	0.83 ^b^
Left-sided neck pain	12 (12.9%)	11 (11.8%)	0.07 ^a^
Right-sided neck pain	20 (21.9%)	9 (9.7%)	
Bilateral neck pain	61 (65.6%)	73 (78.5%)	

Abbreviations: CCT, cervical, cervicothoracic junction, and thoracic spine; UCS, upper cervical spine. ^a^ *p*-value obtained using the Chi-square test. ^b^ *p*-value obtained using the Mann–Whitney U test.

**Table 2 healthcare-12-01348-t002:** Between-group comparison of the mean differences from baseline to post-treatment.

Measures	Group	Baseline	Rosenthal’s r	*p*-Value ^a^	One Week Post-Intervention	Rosenthal’s r	*p*-Value ^a^
Pain intensity (0–10 points)							
	CCT	3.7 ± 2.1	−0.18	0.31	1.8 ± 2.2	−0.17	0.31
	UCS	4.1 ± 2.3			2.2 ± 2.4		
Neck Disability (0–50 points)							
	CCT	22.7 ± 5.4 (45.4%)	0.08	0.14	16.9 ± 3.8 (33.9%)	−0.54	0.01 ^†^
	UCS	22.3 ± 4.4 (44.6%)			19.5 ± 6.8 (38.9%)		
Active CROM (degrees)							
Flexion	CTS	51.2 ± 12.7	−0.06	0.68	51.3 ± 9.7	0.14	0.33
	UCS	52.0 ± 13.0			49.9 ± 10.3		
Extension	CTS	68.6 ± 13.9	0.07	0.60	66.5 ± 14.1	0.07	0.60
	UCS	67.6 ± 12.6			65.5 ± 11.9		
Right lateral flexion	CTS	38.1 ± 7.7	0.15	0.31	40.8 ± 7.5	0.25	0.11
	UCS	36.9 ± 7.6			38.8 ± 9.4		
Left lateral flexion	CTS	43.5 ± 9.1	0.14	0.31	47.5 ± 8.8	0.40	0.03 ^†^
	UCS	42.2 ± 9.2			43.9 ± 9.0		
Right rotation	CTS	64.9 ± 10.5	0.26	0.08	65.6 ± 8.4	0.21	0.11
	UCS	62.4 ± 8.9			63.6 ± 8.6		
Left rotation	CTS	62.1 ± 7.3	0.14	0.32	67.2 ± 6.5	0.02	0.91
	UCS	60.8 ± 11.0			67.06 ± 7.8		

Abbreviations: CCT, cervical, cervicothoracic junction, and thoracic spine; UCS, upper cervical spine. ^a^ *p*-value obtained using the Mann–Whitney U test; ^†^ significant difference.

## Data Availability

For confidentiality purposes, the data are in the possession of the author (R.R.-d.R.).

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
