# Peer review of "Comparison of Short-Term Effects of Different Spinal Manipulations in Patients with Chronic Non-Specific Neck Pain: A Randomized Controlled Trial"

_healthcare, 2024, doi:10.3390/healthcare12131348_

Round 1

Reviewer 1 Report

Comments and Suggestions for Authors

The study's methodology is robust, employing appropriate randomization and blinding techniques to ensure data integrity. The authors conclude that CCT manipulations might offer a slightly more pronounced reduction in disability than UCS manipulations alone, with no significant differences observed in pain intensity or CROM.

2. Major Comments

  1. Introduction and Literature Review:
    • Depth of Review: While the introduction outlines the prevalence and impact of CNNP, it lacks a detailed review of the mechanisms and efficacy of UCS and CCT manipulations. Expanding the literature review to include more recent studies or meta-analyses comparing these techniques could provide a stronger rationale for the study.
    • Research Gap: The research question is relevant; however, the authors could better articulate the unique contributions of this study compared to existing literature, particularly regarding the direct comparison of these two manipulation techniques.
  2. Methodology:
    • Sample Size Justification: The explanation for the sample size is based on statistical power calculations, which is commendable. However, detailing the assumptions such as effect size and variance estimates used for these calculations would strengthen the reader's understanding and the reproducibility of the study.
    • Blinding and Randomization: The description of the blinding process is adequate, but additional details on how the randomization was concealed from the researchers and participants at the time of allocation would enhance the credibility of the blinding process.
  3. Statistical Analysis:
    • Choice of Tests: The manuscript uses non-parametric tests which are appropriate given the non-normal distribution of the data. The authors should provide a brief explanation of why parametric tests were not suitable given the central limit theorem might still apply.
    • Handling of Missing Data: There is no mention of how missing data were handled in the study. Including a discussion on this, especially if there were any dropouts or missing follow-up data, would be informative.
  4. Results:
    • Data Presentation: While the results are presented clearly, the manuscript could improve by including confidence intervals in addition to p-values to quantify the precision of the estimates.
    • Subgroup Analyses: If data permits, subgroup analyses by demographic factors such as age and sex could be insightful, especially if these factors could influence the effectiveness of manipulations.
  5. Discussion and Implications:
    • Contextualization of Results: The discussion provides a good synthesis of findings with the existing literature. However, it could be improved by discussing potential physiological or anatomical reasons for why CCT manipulations might be more effective than UCS manipulations.
    • Limitations: The discussion on limitations is good but could be expanded to include potential biases or confounders that might have influenced the results, such as therapist expertise or patient expectations.
  6. Conclusion:
    • Recommendations for Practice: While the conclusions are appropriately cautious given the results, providing specific recommendations for clinicians regarding when to prefer CCT over UCS manipulations based on the findings could make the conclusions more impactful.
    • Future Research: Suggestions for future research could include longer follow-up periods to assess the long-term effects of these manipulations or trials including other modalities of treatment in combination with manipulations.

3. Minor Comments

  1. Formatting and Grammar:
    • There are several instances where grammatical errors could potentially confuse the reader. A thorough proofread is recommended.
    • Ensure that all figures and tables are referenced in the text and appropriately labeled and described.
  2. References:
    • Some references are quite dated. Where possible, include more recent studies that reflect the current state of research on spinal manipulation.

Comments on the Quality of English Language

Minor changes are requested

Author Response

The study's methodology is robust, employing appropriate randomization and blinding techniques to ensure data integrity. The authors conclude that CCT manipulations might offer a slightly more pronounced reduction in disability than UCS manipulations alone, with no significant differences observed in pain intensity or CROM.

Response: We appreciate the reviewer's acknowledgment of the robust methodology employed in our study, including rigorous randomization and blinding techniques to uphold data integrity. The various comments provided have been addressed, and the necessary text has been added or modified to enhance the clarity of the information presented in the manuscript. The changes have been highlighted in yellow.

  1. Major Comments

Introduction and Literature Review:

Comments 1: Depth of Review: While the introduction outlines the prevalence and impact of CNNP, it lacks a detailed review of the mechanisms and efficacy of UCS and CCT manipulations. Expanding the literature review to include more recent studies or meta-analyses comparing these techniques could provide a stronger rationale for the study.

Response 1: Thank you for the valuable comment. Following the reviewer's recommendations, we have expanded the literature review in our manuscript to include information from several recent systematic reviews and meta-analyses on the mechanisms of action of spinal manipulations (Nim et al., 2021) and the influence of contextual factors on outcomes in patients with musculoskeletal pain (Ezzatvar et al., 2024). These references have been incorporated into the manuscript:

Nim, C. G., Downie, A., O'Neill, S., Kawchuk, G. N., Perle, S. M., & Leboeuf-Yde, C. (2021). The importance of selecting the correct site to apply spinal manipulation when treating spinal pain: Myth or reality? A systematic review. Scientific Reports, 11(1), 23415. https://doi.org/10.1038/s41598-021-02882-z

Ezzatvar, Y., Dueñas, L., Balasch-Bernat, M., Lluch-Girbés, E., & Rossettini, G. (2024). Which Portion of Physiotherapy Treatments' Effect Is Not Attributable to the Specific Effects in People With Musculoskeletal Pain? A Meta-Analysis of Randomized Placebo-Controlled Trials. The Journal of Orthopaedic and Sports Physical Therapy, 54(6), 391–399. https://doi.org/10.2519/jospt.2024.12126

Additionally, information from a systematic review and meta-analysis (Carrasco-Uribarren et al., 2024) has been added, reporting the efficacy of cervical and thoracic spinal manipulation techniques in patients with neck pain. We have also included a recent systematic review and meta-analysis (Liu et al., 2023), which recommends conducting more randomized controlled trials in this field due to the limited number of available studies. All authors believe that the incorporation of these systematic reviews and meta-analyses will significantly enhance the quality of the information provided to justify our study.

Carrasco-Uribarren, A., Pardos-Aguilella, P., Jiménez-Del-Barrio, S., Cabanillas-Barea, S., Pérez-Guillén, S., & Ceballos-Laita, L. (2024). Cervical manipulation versus thoracic or cervicothoracic manipulations for the management of neck pain. A systematic review and meta-analysis. Musculoskeletal science & practice, 71, 102927. https://doi.org/10.1016/j.msksp.2024.102927

Liu, Z., Shi, J., Huang, Y., Zhou, X., Huang, H., Wu, H., Lv, L., & Lv, Z. (2023). A systematic review and meta-analysis of randomized controlled trials of manipulative therapy for patients with chronic neck pain. Complementary Therapies in Clinical Practice, 52, 101751. https://doi.org/10.1016/j.ctcp.2023.101751

Comments 2: Research Gap: The research question is relevant; however, the authors could better articulate the unique contributions of this study compared to existing literature, particularly regarding the direct comparison of these two manipulation techniques.

Response 2: Thank you for this comment. The last paragraph of the introduction has been revised to enhance justification. Regarding the research gap, and based on our understanding, there is currently no randomized clinical trial comparing the effectiveness of these two interventions, which could be beneficial for healthcare professionals when making treatment decisions for patients with CNNP. This information has been added to the text to improve our introduction (lines 85-86).

Methodology:

Comments 3: Sample Size Justification: The explanation for the sample size is based on statistical power calculations, which is commendable. However, detailing the assumptions such as effect size and variance estimates used for these calculations would strengthen the reader's understanding and the reproducibility of the study.

Response 3: Thank you for the comment. An effect size of 2.1 points was considered, the variance used to estimate the mean differences was 2.5, and a significance level of 5% indicated the statistical significance threshold for determining if the observed differences are statistically significant. These assumptions are fundamental for conducting an appropriate and reliable analysis of the differences in NPRS scores. These estimates were based on the study conducted by Cleland, J.A., et al., 2017 [37].

Cleland, J.A.; Childs, J.D.; Whitman, J.M. Psychometric Properties of the Neck Disability Index and Numeric Pain Rating Scale in Patients with Mechanical Neck Pain. Arch Phys Med Rehabil 2008, 89, 69–74, doi:10.1016/J.APMR.2007.08.126.

Comments 4: Blinding and Randomization: The description of the blinding process is adequate, but additional details on how the randomization was concealed from the researchers and participants at the time of allocation would enhance the credibility of the blinding process.

Response 4: We appreciate the reviewer's comment, as it helps clarify the randomization process we employed. A computer-generated random code system was used, created by an independent third party not involved with the research team. However, as acknowledged in the manuscript, due to the nature of the study, blinding the person responsible for administering the spinal manipulations was not possible. Participants were blinded to the study's objective and their group assignment, although they might have inferred the treatment they were receiving during the intervention. Nevertheless, we ensured the blinding of the evaluator and the statistician responsible for data analysis. Additionally, as with most studies involving manual therapy techniques, we recognize the difficulty of blinding the interventionist and participants, which we have noted as a limitation in this study.

Statistical Analysis:

Comments 5: Choice of Tests: The manuscript uses non-parametric tests which are appropriate given the non-normal distribution of the data. The authors should provide a brief explanation of why parametric tests were not suitable given the central limit theorem might still apply.

Response 5: Thank you for the valuable comment. The parametric tests were not suitable in this situation because the data did not meet the essential assumptions required for their application, such as the normality of distribution and homogeneity of variances. Although the central limit theorem suggests that with sufficiently large samples the sampling distribution of the mean tends to be normal, this does not address other parametric assumptions. For instance, if the data exhibit significant skewness or unequal variances, the results of parametric tests may be incorrect. In such cases, it is preferable to use non-parametric tests, which do not depend on these assumptions.

Comments 6: Handling of Missing Data: There is no mention of how missing data were handled in the study. Including a discussion on this, especially if there were any dropouts or missing follow-up data, would be informative.

Response 6: We deeply appreciate your observation regarding the handling of missing data. We are pleased to inform you that the final number of participants was 186. As mentioned in the Flowchart Diagram (figure 3), all variables included in the study were collected for these 186 patients, so there was no missing data. Additionally, there were no patient dropouts during the study period. Thank you again for your valuable suggestion and for your thorough review of our work.

Results:

Comments 7: Data Presentation: While the results are presented clearly, the manuscript could improve by including confidence intervals in addition to p-values to quantify the precision of the estimates.

Response 7: We appreciate the reviewer's comment. While we understand the importance of including confidence intervals to quantify the precision of the estimates, we have found that the table is already quite extensive. Including the confidence intervals in this main table could potentially hinder its interpretation. However, we have included the confidence intervals in the Supplementary Tables 1 and 2, which display the intragroup values for each of the groups (UCS and CCT). In these supplementary tables, the confidence intervals are provided in the column indicating the pre-post score difference for each measurement. Thank you again for your valuable feedback and for helping us improve our manuscript.

Comments 8: Subgroup Analyses: If data permits, subgroup analyses by demographic factors such as age and sex could be insightful, especially if these factors could influence the effectiveness of manipulations.

Response 8:  Following the reviewer's recommendations, we compared the manipulation data based on sex. The results did not show statistically significant differences when comparing the NDI, pain intensity (VAS), and active CROM (flexion, extension, right and left lateral flexion, right and left rotation) between men and women. This information has been incorporated into the manuscript in Section 3, Results (lines 288-289).

Discussion and Implications:

Comments 9: Contextualization of Results: The discussion provides a good synthesis of findings with the existing literature. However, it could be improved by discussing potential physiological or anatomical reasons for why CCT manipulations might be more effective than UCS manipulations.

Response 9: We appreciate this valuable comment from the reviewer. In our study, no statistically significant differences were found between groups in terms of pain intensity and CROM, suggesting that both interventions were equally effective. However, participants who received CCT spinal manipulations showed a slight improvement in self-perceived disability compared to those who received UCS manipulation. As discussed in our article, patient perception of change is crucial in treating these cases. We have added a sentence, with its corresponding reference, to suggest that this difference may stem from the interaction of biomechanical, neurophysiological, and contextual factors (lines 357-359). We acknowledge the significant influence of contextual factors in manual therapy according to the scientific literature. Participants who experienced greater improvement in self-perceived disability received more sessions of spinal manipulation than those in the UCS group, which may have influenced their treatment perceptions. Therefore, we believe it is important to highlight the need to interpret these results cautiously.

Comments 10: Limitations: The discussion on limitations is good but could be expanded to include potential biases or confounders that might have influenced the results, such as therapist expertise or patient expectations.

Response 10: Thank you for this comment. Following the reviewer's recommendations, we have expanded the discussion on the study's limitations. Specifically, we have included information on how the therapist's experience and the patient's expectations could have influenced the reported results (lines 381-385). We believe this addition will enhance the quality of our manuscript.

Conclusion:

Comments 11: Recommendations for Practice: While the conclusions are appropriately cautious given the results, providing specific recommendations for clinicians regarding when to prefer CCT over UCS manipulations based on the findings could make the conclusions more impactful.

Response 11: Thank you for this comment. While our conclusions are cautious, following the reviewer's suggestions, we have added a recommendation for clinical practice based on our findings.

Comments 12: Future Research: Suggestions for future research could include longer follow-up periods to assess the long-term effects of these manipulations or trials including other modalities of treatment in combination with manipulations.

Response 12: We sincerely appreciate this comment, which we believe will enhance recommendations for future research stemming from our study. We have incorporated these suggestions into the text (lines 399-403).

  1. Minor Comments

Formatting and Grammar:

Comments 13: There are several instances where grammatical errors could potentially confuse the reader. A thorough proofread is recommended.

Response 13: We deeply appreciate the reviewer’s comment and apologize for any grammatical errors present in the previous version of the manuscript. We have conducted a thorough review of the document and corrected all identified grammatical errors. These corrections were made with the aim of ensuring clarity and avoiding any potential confusion for the reader.

Comments 14: Ensure that all figures and tables are referenced in the text and appropriately labeled and described.

Response 14: We appreciate the reviewer’s observation. We have carefully reviewed the manuscript to ensure that all figures and tables are referenced in the text and properly labeled and described. We believe they now meet the required standards.

References:

Comments 15: Some references are quite dated. Where possible, include more recent studies that reflect the current state of research on spinal manipulation.

Response 15. We appreciate the reviewer's comments. We have incorporated several recent systematic reviews with meta-analyses and updated some older references with more current studies. We are confident that these changes will significantly improve the representation of the current state of research on spinal manipulation in our manuscript. Thank you again for your thorough review and for helping us improve our work.

Reviewer 2 Report

Comments and Suggestions for Authors

This is an interesting study, although studies with a similar research question have been conducted in the past (middle vs lower cervical spine manips). This study specifically addresses the research question of upper vs lower cervical spine manipulations, although the authors should have first presented the rationale/clinical reasoning for conducting a regional manipulation of the cervical spine. The clinical reasoning is not the same for those two regions.  For instance, the inclusion criterion (3) in lines 85-6 does not clarify if the manip would be region – specific, depending on the cervical spine pain location. This particular study’s strength is the quite large sample size (n=186), although there is no short or longer-term follow up included. Furthermore, the effect of a single treatment session was examined. I think these should be somehow reflected in the title of the manuscript also. Also, why were follow up measurements conducted 1 week later and not perhaps 1 day later? Why did the authors did not provide more than one treatment session? These need to be clarified in detail.

Comments on the Quality of English Language

Adequate

Author Response

Comments 1: This is an interesting study, although studies with a similar research question have been conducted in the past (middle vs lower cervical spine manips). This study specifically addresses the research question of upper vs lower cervical spine manipulations, although the authors should have first presented the rationale/clinical reasoning for conducting a regional manipulation of the cervical spine. The clinical reasoning is not the same for those two regions. 

Response 1: We greatly appreciate the reviewer's comments and the time taken to thoroughly review our manuscript. We agree with the reviewer that previous studies have investigated similar research questions, specifically comparing the effectiveness of mid-cervical spine manipulations versus lower cervical or thoracic spine manipulations. However, a recent systematic review and meta-analysis (Liu et al., 2023) suggested that the number of high-quality methodological studies in this field remains limited, highlighting the need for more randomized controlled trials. This information, along with the corresponding reference, has been added to the text to enhance the justification for our study (lines 67-70).

Liu, Z., Shi, J., Huang, Y., Zhou, X., Huang, H., Wu, H., Lv, L., & Lv, Z. (2023). A systematic review and meta-analysis of randomized controlled trials of manipulative therapy for patients with chronic neck pain. Complementary therapies in clinical practice, 52, 101751. https://doi.org/10.1016/j.ctcp.2023.101751

Furthermore, we have restructured the last paragraph of the introduction, dividing it into two separate paragraphs to improve the study's justification and provide greater clarity for the reader.

Comments 2:  For instance, the inclusion criterion (3) in lines 85-6 does not clarify if the manip would be region – specific, depending on the cervical spine pain location. 

Response 2: Thank you for your insightful comment, which we sincerely appreciate. We acknowledge that this detail was not explicitly stated in the text. The study's inclusion criteria were designed to identify patients with chronic nonspecific neck pain, characterized by localized pain in the cervical region triggered by repetitive movements or sustained postures lasting more than 12 weeks. Spinal manipulation techniques were uniformly applied to all patients, irrespective of the specific location of cervical pain, as a recent systematic review indicated no differences in applying spinal manipulations over a clinically relevant versus non-relevant cervical region.

Nim, C.G.; Downie, A.; O’Neill, S.; Kawchuk, G.N.; Perle, S.M.; Leboeuf-Yde, C. The Importance of Selecting the Correct Site to Apply Spinal Manipulation When Treating Spinal Pain: Myth or Reality? A Systematic Review. Scientific Reports 2021 11:1 2021, 11, 1–13, doi:10.1038/s41598-021-02882-z.

As detailed in subsection "2.4. Interventions," spinal manipulation techniques were performed bilaterally. Nevertheless, for clarity, we have included an additional sentence in this subsection indicating that these techniques were uniformly applied, regardless of the cervical region affected by pain (lines 137-139). Furthermore, we have incorporated the mentioned reference into our manuscript.

Comments 3: This particular study’s strength is the quite large sample size (n=186), although there is no short or longer-term follow up included. Furthermore, the effect of a single treatment session was examined. I think these should be somehow reflected in the title of the manuscript also. Also, why were follow up measurements conducted 1 week later and not perhaps 1 day later? Why did the authors did not provide more than one treatment session? These need to be clarified in detail.

Response 3: We deeply appreciate the reviewer’s comment. We have modified the manuscript title for greater clarity, emphasizing that this study compared the short-term effects of both interventions.

Furthermore, the present study aimed to compare the short-term effects of UCS manipulation versus a combination of CCT spine manipulations on pain intensity, disability, and CROM in patients with CNNP. It was designed to evaluate the effectiveness of a single session of spinal manipulations on these outcome measures.

Neck disability, measured using the Neck Disability Index (NDI), was one of our primary outcome measures. The NDI is a 10-item questionnaire related to the patient’s daily activities (pain intensity, personal care, lifting, reading, headaches, concentration, work, driving, sleeping, and recreation). There are studies, included in the discussion, that apply a single session of spinal manipulations and remeasure cervical disability two (Dunning et al., 2012) or seven days (Saavedra et al., 2013) after the intervention. Similar to Saavedra et al., 2013, we decided to conduct follow-up measurements one week later to allow patients to complete this questionnaire with perspective, based on how they felt during the week following the intervention.

We appreciate your suggestion to provide a clearer explanation of the timing and rationale for the follow-up measurements and the decision to use a single treatment session. This has been clarified in the manuscript to ensure comprehensive understanding (lines 186-189). Thank you for your valuable input, which has undoubtedly improved the quality of our work.

Reviewer 3 Report

Comments and Suggestions for Authors

I appreciate the invitation to review this manuscript. It is a study with a robust and very clear methodology. I have a few minor comments that I would like to clarify before recommending publication of this manuscript.

1. The authors do a good job in the introduction of justifying the need for this study. Apropos contextual factors, the authors could elaborate on this point by reviewing a recent study published in JOSPT (10.2519/jospt.2024.12126) on the effect of physical therapy interventions, including manual therapy. 

2. The authors describe the exclusion criteria well. However, I would like them to be clearer about whether a clinical assessment of these contractions was performed, e.g., was the alar ligament test, or the vertebral artery test, assessed?

3. The discussion phrase "suggesting that this effect may be more closely linked to the neurophysiological aspects of spinal manipulation than its mechanical effects [38]." (line 315-317) is key to the rationale for this study. Therefore, I recommend that this also be mentioned in the introduction. 

4. The authors also well acknowledge the limitations of the study. Great job.

5. I would recommend to the authors that they add a sentence about in the conclusion of this study on recommendations for clinical practice, e.g., the positive short-term effects of manual therapy can be used in combination with other active therapies such as exercise and education to optimize outcomes for patients with persistent neck pain. In addition, highlight the role that contextual factors may play in manual therapy.

Author Response

I appreciate the invitation to review this manuscript. It is a study with a robust and very clear methodology. I have a few minor comments that I would like to clarify before recommending publication of this manuscript.

Response: Dear reviewer, thank you for your thoughtful review of our randomized controlled trial. We value your comments, questions, and concerns, as they play a vital role in improving the quality and applicability of our manuscript. We have carefully addressed each of your points, and the revised manuscript highlights these changes in yellow. We trust that these revisions will enhance the manuscript.

Comments 1: The authors do a good job in the introduction of justifying the need for this study. Apropos contextual factors, the authors could elaborate on this point by reviewing a recent study published in JOSPT (10.2519/jospt.2024.12126) on the effect of physical therapy interventions, including manual therapy.

Response 1: We deeply appreciate this comment, as we were unaware of the existence of this recent systematic review with meta-analysis. We agree that including this systematic review will help improve the quality of our introduction. Additionally, as the recommended study indicates, enhancing these contextual factors is an ethical way to achieve greater benefits for patients. This information has also been added to the text (lines 67-70).

Comments 2: The authors describe the exclusion criteria well. However, I would like them to be clearer about whether a clinical assessment of these contractions was performed, e.g., was the alar ligament test, or the vertebral artery test, assessed?

Response 2: We sincerely appreciate the reviewer's comment. Prior to proceeding with cervical spinal manipulations, participants underwent a comprehensive clinical evaluation to rule out red flags such as vertebrobasilar insufficiency and upper cervical spine instability. In accordance with the reviewer's recommendations, we have incorporated this information into the text (lines 110-113) within subsection "2.2. Participants". Additionally, we have included two additional references supporting this clinical assessment. 

Hutting, N.; Scholten-Peeters, G.G.M.; Vijverman, V.; Keesenberg, M.D.M.; Verhagen, A.P. Diagnostic Accuracy of Upper Cervical Spine Instability Tests: A Systematic Review. Phys Ther 2013, 93, 1686–1695, doi:10.2522/PTJ.20130186.

Hutting, N.; Verhagen, A.P.; Vijverman, V.; Keesenberg, M.D.M.; Dixon, G.; Scholten-Peeters, G.G.M. Diagnostic Accuracy of Premanipulative Vertebrobasilar Insufficiency Tests: A Systematic Review. Man Ther 2013, 18, 177–182, doi:10.1016/J.MATH.2012.09.009.

We are confident that the inclusion of this suggestion will significantly enhance the quality of our manuscript.

Comments 3: The discussion phrase "suggesting that this effect may be more closely linked to the neurophysiological aspects of spinal manipulation than its mechanical effects [38]." (line 315-317) is key to the rationale for this study. Therefore, I recommend that this also be mentioned in the introduction. 

Response 3: Thank you for this comment. We agree that explaining the possible mechanisms of action of spinal manipulation techniques would enhance the information provided in the introduction. Therefore, following the reviewer's recommendations, we have included these possible mechanisms in the text (lines 54-62).

Comments 4: The authors also well acknowledge the limitations of the study. Great job.

Response 4: We sincerely appreciate your positive comment regarding our acknowledgment of the study's limitations. We value your recognition and appreciate your contribution to improving our work.

Comments 5: I would recommend to the authors that they add a sentence about in the conclusion of this study on recommendations for clinical practice, e.g., the positive short-term effects of manual therapy can be used in combination with other active therapies such as exercise and education to optimize outcomes for patients with persistent neck pain. In addition, highlight the role that contextual factors may play in manual therapy.

Response 5: We deeply appreciate your valuable recommendations. We have accepted your suggestion and incorporated the recommendation for future studies into the conclusion of our paper (lines 399-403). Thank you for your consideration and contributions, which have undoubtedly enriched our work.

Reviewer 4 Report

Comments and Suggestions for Authors

Dear authors, the article the article you present, “Comparison of different spinal manipulations in patients with 2 chronic non-specific neck pain: a randomized controlled trial “is of interest to the scientific community and health professionals, however it needs to clarify/made more objective some things, namely:

1-In the abstract the objective of the study is not explicitly presented. It must be explained;

2-It is not clear how they controlled the variables. They must explore better;

3-It is not clear how the patients were drawn. It must be explained;

4-Line 153 It should be more clear when pain and disability were assessed, as it is not clear;

5-Table 2 is not clear when the post-treatment was, was it a week after? The reason for this option must be justified

Author Response

Dear authors, the article the article you present, “Comparison of different spinal manipulations in patients with 2 chronic non-specific neck pain: a randomized controlled trial “is of interest to the scientific community and health professionals, however it needs to clarify/made more objective some things, namely:

Response: Dear Reviewer, thank you for taking the time to review our randomized controlled trial. We appreciate all your comments, questions, and concerns, as they are crucial for enhancing the quality and generalizability of our manuscript. We have addressed all your concerns thoroughly, and changes have been highlighted in yellow in the revised version of our manuscript. We hope that these revisions improve the manuscript.

Comments 1: In the abstract the objective of the study is not explicitly presented. It must be explained.

Response 1: Thank you for your valuable comments. We have reviewed and updated the abstract to explicitly include the objective of the study as per your suggestion (lines 14-16). We appreciate your observation and are confident that this modification enhances the clarity and precision of our abstract.

Comments 2: It is not clear how they controlled the variables. They must explore better.

Response 2: We are very grateful for this comment, as we believe it will significantly enhance the clarity of our study. To clarify the control of variables, we have included a new subsection titled "2.3. Data Collection," where we have separated the information regarding the measurement of outcome measures from the interventions performed (lines 98-107).

Comments 3: It is not clear how the patients were drawn. It must be explained;

Response 3: We appreciate this comment, which we believe will help improve the quality of the study methodology presentation. Advertising was conducted through posters and social media to achieve the required sample size. Interested individuals with cervical pain attended the private physiotherapy clinic where the clinical trial took place. There, these individuals were subjected to eligibility criteria to determine their inclusion in the study. We have added this information to the "2.2. Participants" subsection (lines 104-105).

Comments 4: Line 153. It should be more clear when pain and disability were assessed, as it is not clear.

Response 4: We sincerely appreciate your comment and thorough review. We have clarified the timing of pain intensity and disability assessments within the section titled "2.5. Outcomes measure" (lines 186-189). These measurements are now clearly detailed, both at baseline and one week post-intervention, following the methodology of a previously published study (Saavedra et al., 2013).

Saavedra-Hernández, M.; Arroyo-Morales, M.; Cantarero-Villanueva, I.; Fernández-Lao, C.; Castro-Sánchez, A.M.; Puentedura, E.J.; Fernández-de-las-Peñas, C. Short-Term Effects of Spinal Thrust Joint Manipulation in Patients with Chronic Neck Pain: A Randomized Clinical Trial. Clin Rehabil 2013, 27, 504–512, doi:10.1177/0269215512464501.

Your suggestions have been instrumental in enhancing the clarity and precision of our manuscript, thereby contributing to its overall quality of presentation. Thank you once again for your constructive feedback.

Comments 5: Table 2 is not clear when the post-treatment was, was it a week after? The reason for this option must be justified.

Response 5: Thank you sincerely for your comment and detailed review. We have clarified in Table 2 that the post-treatment assessment was conducted one week after the intervention, following the methodology of the aforementioned study, which is similar to the present study. This clarification accurately reflects the timing of the post-intervention assessment, ensuring methodological consistency and comparability with previous studies. We appreciate once again your constructive feedback.

Round 2

Reviewer 1 Report

Comments and Suggestions for Authors

I would like to extend my heartfelt congratulations on the outstanding final result of your manuscript.

Reviewer 2 Report

Comments and Suggestions for Authors

The manuscript has now clearly improved and can proceed to publication. The overall merit of the manuscript is judged as average, only because the relative effect of a single session manipulation (applied in 2 different spinal regions) was examined with no long term follow up. However the research question answered is significant, taking into account the limitations of this study.